# “Got Milk Alternatives?” Understanding Key Factors Determining U.S. Consumers’ Willingness to Pay for Plant-Based Milk Alternatives

**DOI:** 10.3390/foods12061277

**Published:** 2023-03-17

**Authors:** Meike Rombach, David L. Dean, Vera Bitsch

**Affiliations:** 1Department of Land Management and Systems, Lincoln University, Lincoln 7647, New Zealand; 2Department of Agribusiness and Markets, Lincoln University, Lincoln 7647, New Zealand; 3School of Management and School of Life Sciences, Chair of Economics of Horticulture and Landscaping, Technical University of Munich, 85354 Freising, Germany

**Keywords:** plant-based milk alternatives, animal welfare, food curiosity, PLS-SEM, climate impact, sustainability

## Abstract

Milk is an important dairy product in U.S. food retail. Lifestyle changes toward climate-conscious consumption, animal welfare, and food safety concerns have increased the popularity of plant-based milk alternatives. This study is focused on such beverages and provides insights and best practice recommendations for marketing managers in the U.S. food retail sector. An online survey was distributed to explore factors explaining the intentions of U.S. consumers to purchase and pay a premium for plant-based milk alternatives. Food curiosity and food price inflation were identified as relevant for both willingness to buy and willingness to pay a price premium. In addition, animal welfare concerns and the green and clean product image of plant-based alternatives were relevant to the willingness to pay a premium for plant-based milk.

## 1. Introduction

Milk is a popular product in U.S. consumer markets [1,2,3]. However, consumer awareness and lifestyle changes toward climate-conscious consumption have increased over the past decade [4,5]. This has contributed to plant-based milk alternatives becoming popular replacements for traditional dairy products, such as regular milk, often because they are seen as more environmentally friendly and ethical options [6,7]. Plant-based milk alternatives are beverages made from plant water extracts that mimic the color and creaminess of regular milk [8,9,10]. Such beverages are marketed as “milk” in the U.S. despite consisting of plant materials [11,12,13].

In addition to those who only eat plant-based foods, consumers with dietary restrictions also prefer these beverages [14,15]. Consumers with dietary restrictions due to allergies, intolerance, and hypercholesterolemia buy plant-based milk alternatives [16,17,18,19], while others are motivated by human health, animal welfare, and climate concerns [20]. Health concerns are related to the transmission of zoonoses and antibiotic-resistant pathogens, which can be transmitted to humans through the ingestion of dairy milk [21]. Negative environmental impacts include water depletion, disruption of nutrient cycles, and emissions attributed to milk production [22]. In addition, some consumers have concerns about animal husbandry conditions and production practices [23]. Recent studies highlight an increased product assortment of plant-based milk alternatives [24,25]. In the U.S., five different beverage types are available, classified as grain-, legume-, nut-, seed-, and pseudo-grain-based products [26,27]. The per capita consumption of plant-based milk alternatives was 2.7 kg in 2022, an increase of 1.2 kg since 2013 [27]. In 2022, almond, oat, and soymilk alternatives were the most purchased options among U.S. consumers. Pea milk is an upcoming product that is increasing in popularity [27,28]. Plant-based milk is consumed in higher quantities than other plant-based products sold by U.S. food retailers, such as meat and pre-cooked meals [27].

Against this backdrop, it is surprising that the recent body of literature on consumer behavior and plant-based milk alternatives is not as comprehensive as research on meat alternatives [24]. Several studies have explored alternative meat products in terms of consumer perception, attitudes, and willingness to try, buy, and pay a price premium [26,29,30,31,32]. A few studies have compared different product types, such as dairy, meat, and fish, or have specifically addressed the consumer perspective on plant-based meat alternatives [24]. The recent body of literature on plant-based milk alternatives has covered brand image, buying motivation, and attribute preferences [19,20,33], consumer segmentation studies [24], and studies related to the health and beverage choices of adults and children. This study aims to add to the recent body of literature and explore drivers and inhibitors of consumer willingness to pay (WTP) and their willingness to pay more (WTPM) for plant-based milk alternatives. The body of literature related to plant-based milk alternatives is still rather limited, but it can be assumed that consumer motivation, perception, and attitudes are similar to other plant-based food substitutes [9,24]. Potentially relevant factors, hypotheses, and a proposed conceptual model are presented in the following section of the paper.

## 2. Literature Review and Conceptual Model

### 2.1. Animal Welfare Concerns

Animal welfare has become crucial in purchasing decisions for meat and dairy products [34,35]. Animal welfare certification helps to ease consumer concerns by alleviating information asymmetry [36]. In particular, consumers demand information about on-farm welfare [37,38], which includes species-appropriate husbandry, feed, insemination, and sanitary practices, as well as ethical and painless slaughtering practices [36,39,40,41]. With animal welfare as one of the main reasons consumers refrain from dairy product consumption, the following hypotheses are proposed:

**Hypothesis** **(H1).***Animal welfare concerns positively impact consumers’ willingness to pay for plant-based milk alternatives*.

**Hypothesis** **(H2).***Animal welfare concerns positively impact consumers’ willingness to pay more for plant-based milk alternatives*.

### 2.2. Food Curiosity

While animal welfare as a critical factor determining consumer intention to buy plant-based food products and other novel foods has been well investigated, food curiosity is another important factor that has not been as widely explored to date [42,43]. Food curiosity refers to an enticing feeling that drives consumers to explore food. This includes food production, processing, and consumption [44]. The feeling is characterized by consumers’ awareness of food-related knowledge gaps and the urge to close those gaps through information seeking [43]. Food curiosity is an exploratory behavior and a key driver that is a suitable predictor of acceptance and willingness to try, buy, and pay a price premium for plant-based food alternatives [7,26,45]. Drawing from the body of literature on meat alternatives in the absence of studies on milk alternatives, the following hypotheses are proposed:

**Hypothesis** **(H3).***Food curiosity positively impacts consumers’ willingness to pay for plant-based milk alternatives*.

**Hypothesis** **(H4).***Food curiosity positively impacts consumers’ willingness to pay more for plant-based milk alternatives*.

### 2.3. Food Safety

Milk value chain actors, such as food retailers, processors, and producers, must act immediately after a food recall, as food safety is considered a fundamental right for all consumers in most countries. The U.S. has a high baseline standard for food safety, where the Food and Drug Administration (FDA) oversees milk safety programs [46]. In terms of food safety and milk, consumers appreciate plant-based milk alternatives because dairy-based beverages expose them to the risk of zoonosis, mycotoxins, antibiotic-resistant pathogens, and growth hormones, such as recombinant bovine somatotropin (rbST) [47,48,49,50,51,52]. Consumers voice their concerns about hygiene, cleanliness, and food safety in public debates, and are willing to pay price premiums for food safety as a product attribute [47]. Therefore, the following hypotheses are proposed:

**Hypothesis** **(H5).***Food safety concerns positively impact consumers’ willingness to pay for plant-based milk alternatives*.

**Hypothesis** **(H6).***Food safety concerns positively impact consumers’ willingness to pay more for plant-based milk alternatives*.

### 2.4. Green and Clean Image of Plant-Based Milk Alternatives

Agricultural production is scrutinized for its potential adverse impact on the environment. Soil degradation, greenhouse gases, water pollution, and negative effects on biodiversity are attributed to animal husbandry and, consequently, milk production [20]. These impacts can lead to consumers feeling conflicted about their consumption habits. While consumers may disapprove of the environmental effects, they may enjoy consuming milk [53]. The occurrence of both positive and negative perceptions leads to inconsistent attitudes toward the product. These unanswered questions can lead consumers to follow dairy-free diets and lean toward plant-based milk alternatives [53]. In line with discussions on conscious consumption, food security, and justice, alternative food products not involving animal husbandry have been marketed as more ethical, environmentally friendly, and sustainably produced options [54]. In this context, the following hypotheses are proposed:

**Hypothesis** **(H7).***The green and clean image of plant-based milk alternatives positively impacts consumers’ willingness to pay for plant-based milk alternatives*.

**Hypothesis** **(H8).***The green and clean image of plant-based milk alternatives positively impacts consumers’ willingness to pay more for plant-based milk alternatives*.

### 2.5. Impact of Food Price Inflation

COVID-19 caused significant disruptions in food supply chains. Increased unemployment, food price inflation, and the global effects of the Russia–Ukraine war have further impacted consumer behavior [55,56,57]. Reportedly, consumers are prioritizing grocery shopping over buying fast food [55], and the way U.S. consumers purchase groceries has shifted to contactless buying in the wake of COVID-19 [57]. In addition to changes toward more basic and cheaper products, consumers have employed shopping strategies, such as bulk buying, to save money [58], and have fallen prey to panic-buying sprees as they felt overwhelmed by the situation [59]. Given the higher price point of plant-based milk alternatives and the fact that consumers are adjusting their buying behavior to higher prices [60,61], the following hypotheses are proposed:

**Hypothesis** **(H9).**
*Food price inflation negatively impacts consumers’ willingness to pay for plant-based milk alternatives.*


**Hypothesis** **(H10)****.**
*Food price inflation negatively impacts consumers’ willingness to pay more for plant-based milk alternatives.*


The proposed conceptual model is grounded in the literature discussed (Figure 1). It is suggested that U.S. consumers’ willingness to pay and their willingness to pay more for plant-based milk alternatives is influenced by attitudes, concerns, and feelings of the consumers, perceptions of product characteristics, and external economic factors.

## 3. Materials and Methods

### 3.1. Survey Instrument, Sampling, and Recruitment

The data for this study were collected through Qualtrics XM (Washington, USA), an online survey software. Survey participants had to be at least 18 years of age, reside in the U.S., be in charge of household shopping, and have an interest or experience in plant-based milk alternatives. The survey link was disseminated via Amazon Mechanical Turk (Mturk) in December 2022. Mturk is a platform that has operated since 2005 and is widely used by academics in management, psychology, marketing, and economics research [62]. The platform connects researchers with registered workers who are willing to complete surveys and online experiments [63,64]. The survey instrument was pre-tested by fifteen workers on Mturk. Pre-testing is necessary for an optimized setup; for instance, ensuring the clarity of the survey items and instructions and reducing respondent frustration [65]. For this study, the pre-test allowed the researchers to more accurately determine the estimated completion time and to adjust the phrasing of questions that workers found difficult to understand.

The survey consisted of questions on consumers’ socio-demographic backgrounds, attitudes, concerns regarding animal welfare and food safety, and feelings, including food curiosity and perception of plant-based milk alternatives’ product image. In addition, questions about the impact of food price inflation on their shopping behavior were asked, as well as respondents’ willingness to pay and to pay a price premium for plant-based milk alternatives. The items were derived from the recent body of literature [44,53,55,60] and adjusted to the study context. All questions were closed-end questions asking consumers to indicate their agreement on a seven-point Likert scale.

Initially, 500 survey responses were obtained; however, 14 had to be excluded for incompleteness and speeding behavior [66] with a response time far below the average time of 15 min. The sample size of 486 U.S. residents was considered satisfactory in exploring the main factors driving their willingness to pay and their willingness to pay more for plant-based milk alternatives via partial least squares structural equation modeling (PLS-SEM). Hair et al. (2022) emphasize the “10-times rule” to determine the minimum sample size; effectively, 10× the largest number of model links pointing at any latent variable [67].

### 3.2. Analysis

Two software packages were used for the analysis: descriptive statistics were generated with SPSS 28 (IBM, New York, NY, USA), and SmartPLS 4 ( SmartPLS GmbH, Oststeinbek, Germany) used for the PLS-SEM analysis. PLS-SEM performs a series of simultaneous PLS regressions and is particularly appropriate for estimating complex models [67,68,69]. The PLS-SEM approach follows a twofold procedure by assessing the measurement model (outer model analysis) first and then the structural model (inner model analysis) [69].

The outer model evaluation performs reliability and validity checks. According to Hair et al. (2022), this entails an assessment of item/scale factor loadings, Cronbach’s alpha (Cr.A), composite reliability (C.R.), and the average variance extracted (AVE) [67]. For reliability, both Cr.A and the C.R. scores should be 0.6 or greater [67,69]. For convergent validity of exploratory models, Hair et al. (2022) suggest that item/scale factor loadings should be above 0.4 [67] and scale AVE values should equal or exceed 0.6 [67]. To test discriminant validity, both the heterotrait–monotrait ratio of correlations criterion (HTMT) and the Fornell–Larcker criterion were employed [67,70,71]. To satisfy the Fornell–Larcker criterion, the square root of a construct’s AVE should exceed any cross-scale correlations [67,70]. The HTMT is satisfied if values are less than 0.9 [69]. Finally, variance inflation factor (VIF) scores of 5 or greater indicate that multicollinearity could be a problem in the model, so a VIF of less than 5 is the target [67].

After completing the analysis of the outer (measurement) model, the inner (structural) model was evaluated, followed by hypothesis testing. The outer model analysis required bootstrapping with 5000 test samples [67]. Bootstrapping examines these test samples to test the significance of the estimated path coefficients. Further, 5 model performance checks were performed, including overall goodness of fit (GoF) and normed fit index (NFI), which are better when larger; standardized root mean square residual (SRMR), which is good when less than 0.08 and problematic if larger than 0.1; explanatory power (R^2^), which is small, moderate, and large if near 0.25, 0.5, and 0.75, respectively; and predictive relevance (Q^2^), which is acceptable, medium, and strong if greater than 0, 0.25, and 0.5, respectively [67,69].

## 4. Results and Discussion

### 4.1. Survey Participants’ Socio-Demographic Backgrounds

Table 1 displays the socio-demographic information of the survey participants. In the sample, 49% of the participants identified as male and 51% as female. Most respondents were between 25–44 years old and had earned a bachelor’s degree or higher. Their annual pre-tax household income ranged from $25,000 to $75,000. Regarding geographical location, 47.3% of participants resided in the South, 21.6% in the Northeast, 17% in the Midwest, and 13.4% in the West of the United States. Compared with the U.S. population, the sample can be described as younger and more educated, and mostly in the low- to mid-range income bracket.

### 4.2. Measurement Model Results

As displayed in Table 2, the Cronbach’s alpha and composite reliability indicators were above 0.6. This indicates that they exceeded the threshold affirming construct reliability. Similarly, the average variance extracted (AVE) values were above 0.5, and factor loadings of all items were above 0.6. Respectively, the composite reliability values indicate good internal consistency reliability, and all latent variables fulfilled the threshold value and were therefore considered to fulfill the standard recommendation for convergent validity. Moreover, discriminant validity requirements were achieved for all constructs, as shown in Table 3. There were no discriminant validity issues with satisfactory results in both HTMT ratios and Fornell–Larcker criterion scores [67,70]. The VIF values were acceptable, ranging from 1.187 to 2.195, so there was no evidence of model multicollinearity [70].

### 4.3. Structural Model Results

The structural model was assessed for its goodness of fit, explanatory power, and predictive relevance. The model had a normal fit index (NFI) of 0.751, a standardized root mean square residual (SRMR) of 0.086, and an overall goodness of fit (GoF) of 0.637, indicating an adequate model fit.

The model’s explanatory power was moderate, with the model’s constructs contributing to an R^2^ of 0.541 or 54.1% of the variance for willingness to pay, and 0.625 or 62.5% of the variance for willingness to pay more for plant-based milk alternatives. This suggests that the model can explain consumer behavior of willingness to pay (moderate commitment) and willingness to pay a premium (higher commitment). Predictive relevance was tested using the Stone–Geisser criterion Q^2^. Given that all Q^2^ values were higher than zero, the model has adequate predictive relevance. The average Q^2^ score of 0.564 indicates strong predictive relevance.

### 4.4. Hypothesis Testing Results

Animal welfare concerns did not significantly impact willingness to pay for plant-based milk alternatives (H1), but they did significantly impact willingness to pay a premium for plant-based milk alternatives (H2). Additionally, H3 and H4 found support as food curiosity significantly impacts U.S. consumers’ willingness to pay and their willingness to pay more for plant-based milk alternatives. Food safety concerns significantly impact willingness to pay for plant-based milk alternatives (H5) but did not influence willingness to pay a price premium for plant-based milk alternatives (H6). A significant relationship was found between the green and clean image and willingness to pay a price premium (H8), but no significant relationship was found between the green and clean image and willingness to pay (H7). Lastly, H9 and H10 both found support as food price inflation has a significant impact on willingness to buy and pay a price premium for plant-based milk alternatives (Table 4 and Figure 2).

## 5. Discussion

This study was dedicated to key factors relevant to predicting U.S. consumers’ willingness to pay and their willingness to pay more for plant-based milk alternatives. Overall, the conceptual model, which was based on the analysis of prior work on both dairy milk and meat alternatives, was satisfactorily fit, had moderate explanatory power, and had strong predictive relevance. The results emphasize food curiosity and food price inflation for both forms of consumer behavior, while animal welfare concerns and green and clean product image were only relevant for the willingness to pay a price premium.

The results are well in line with the recent body of literature. Overall, food curiosity is a predictor that leads consumers to evaluate food products more favorably and is a good predictor for willingness to try novel plant-based food products, including plant-based milk alternatives [43]. Some studies on novel plant-based food alternatives and other novel food products have shown the impact of curiosity on willingness to buy and pay any price premium [72,73,74]. Studies dedicated to plant-based meat alternatives highlight that traditional meat-eaters do not intend to reduce meat consumption but appreciate technologies that allow for healthier eating. The same may hold for milk consumption. It can be expected that consumers appreciate alternative milk alternatives that are new and innovative as they satisfy their curiosity [44].

The results related to food price inflation can be explained by the fact that U.S. consumers are adjusting their buying behavior to high prices [55,60]. It was found that consumers are more price sensitive than in previous years and tend to buy generic products over brands. This holds particularly true for consumers with low incomes [60]. Food price inflation has impacted low-income consumers as food prices for healthier and stable food products, including milk and milk alternatives, are increasingly more expensive than unhealthy options [50,60,75]. As a specific predictor for willingness to pay and willingness to pay for plant-based food and beverages, food price inflation is yet to be more widely explored. Overall, studies examining this in the U.S. context are rare. However, it should be noted that irrespective of their brand, plant-based milk alternatives have a relatively high price point, which may have impacted the results of this study. U.S. consumers’ willingness to pay a price premium for plant-based options refers to a consumer’s willingness to pay more for a competing product [76]. This may be a traditional milk product or other plant-based products substituting milk. Reasons to pay more for plant-based milk alternatives are animal welfare concerns and the green and clean image of the product, suggesting environmental friendliness and sustainability [15,20]. Regardless, the price premium is often associated with consumers’ perception of the brand and quality [76]. Perceived quality is influenced by the sensory profile of plant-based milk alternatives. Taste, texture, and flavor influence willingness to buy and pay a price premium. Marketing and sensory studies emphasize that the taste of plant-based milk products can be an inhibitor to consumer acceptance and, ultimately, repeat purchases and willingness to pay [16,77]. The soluble fibers in plant-based milk influence the mouthfeel and texture of the product, which is commonly appreciated by consumers. Conversely, plant-based milk products with a strong presence of plant phenols and bioactive compounds, such as isoflavonoids, have been rejected by consumers due to their resultant aftertaste. Furthermore, negative experiences related to past purchases, e.g., products of inferior quality, tend to inhibit consumers’ willingness to pay [77]. The green and clean product image may also contribute to these factors. A recent study on traditional and plant-based meat emphasizes that green and clean image is a significant predictor; however, it highlights an inelasticity for willingness to pay more [78]. Overall, the recent body of literature appears inconclusive on whether or not a “green and clean image” is a positive predictor for willingness to pay or willingness to pay more for plant-based foods and beverages. Following Boukid (2021), depending on the study context, the factor has been a driver and an inhibitor [79].

## 6. Conclusions

### 6.1. Suggestions for Practitioners

The findings of this study provide insights for marketing managers in U.S. food retail chains and gastronomy. The results regarding consumer attitudes and concerns can be used to target different kinds of consumers buying plant-based milk alternatives. Emphasizing animal welfare will appeal to vegans and other ethically conscious consumers. The green and clean product image may be equally important to these consumers, as well as those who are climate conscious. However, marketers must consider which aspects are emphasized to advertise plant-based milk alternatives. For instance, nut-based beverages comprising Californian almonds should not be blindly praised for being environment-friendly, as the product has a high water footprint [20]. In this case, a comparison with regular milk may be relevant [20]. U.S. consumers buying both regular milk and plant based-milk alternatives and price-conscious consumers will require specific attention, as the impact of food price inflation may lead to switching back to the regular product.

Marketers can use social media platforms to trigger food curiosity for plant-based milk alternatives. Social media platforms allow customers to share their experiences, and food is one of the topics that users like to comment on or even share user-generated content about, such as photos and videos [80]. Studies have found that engagement through the combination of image and commentary extends the user experience beyond the moment of interaction and allows them to recognize the content and product later on [80].

### 6.2. Limitations and Suggestions for Future Research

As the data from this study were procured via Mturk, a crowd-sourcing platform, critical reflection is required. A sample from this crowd-sourcing platform may not be representative of the U.S. population. The sample is, however, more representative of the overall U.S. population than an online convenience sample, or a sample of university students [64]. In addition, the sample of this study under-represents older and less educated consumers, so the generalizability of the results to 65-and-older consumers is somewhat limited. Even though the main consumers of plant-based milk alternatives tend to be millennials and gen z, the voices of the elder generation should not be missed [6,14,81]. Therefore, in future research, recruitment through an opt-in panel provider and quota sampling will allow for increasing the likelihood of the sample representing the U.S. population. Future studies could compare various types of plant-based milk alternatives and frame the work in a sustainability context. Current work largely emphasizes environmental sustainability, e.g., low emissions, and neglects the economic and social pillars of sustainability. Investigating hemp and pea milk appears to be promising in this context. These products are, in a consumer context, not yet widely researched, even though they are long-established alternative milk options. While pea milk is increasing its market share steadily, hemp milk appears to be still a niche product in U.S. food retail. The authors of this study acknowledge a limitation to their survey design concerning pea and hemp milk alternatives, which have been used as examples of “new” plant-based milk in some of the food curiosity questions. While pea milk is relatively new, it may be too popular to be suitable as an example for a food curiosity question. However, both products are relatively new and have substantially lower market shares compared to almond, oat, and soymilk, so perhaps they can be considered appropriate at the time of data collection. A further limitation of this study is the fact that sensory attributes, such as appearance, texture, taste, and flavor, were not included as predictors for willingness to buy and pay a price premium. In future studies, a combination of sensory evaluation, hypothetical choice experiments, or experimental auction may help to overcome this limitation. Such a combination allows consumers to evaluate the sensory and commercial attributes of the product. Further studies could also try to recruit consumers who drink both plant-based milk alternatives and regular dairy milk. The investigation could be framed in the context of value promises and brand switching or exploring preferences for these products in varying consumption situations.

## Figures and Tables

**Figure 1 foods-12-01277-f001:**
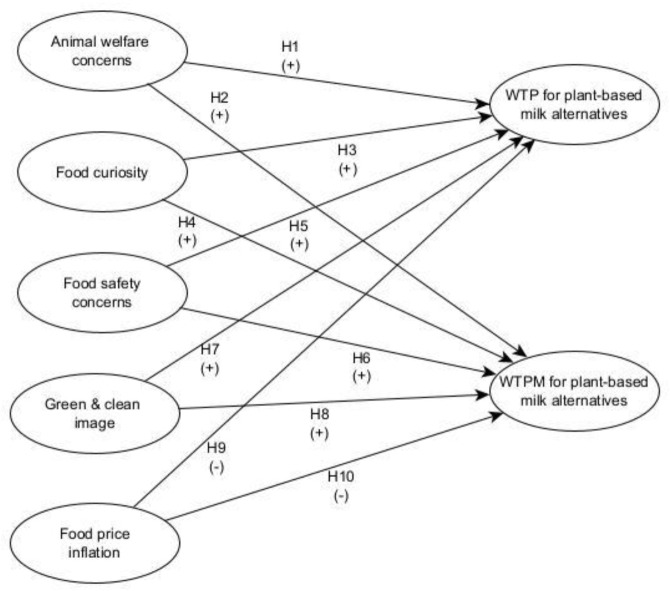
Proposed conceptual model. Note: Willingness to pay is abbreviated as WTP; Willingness to pay more is abbreviated as WTPM.

**Figure 2 foods-12-01277-f002:**
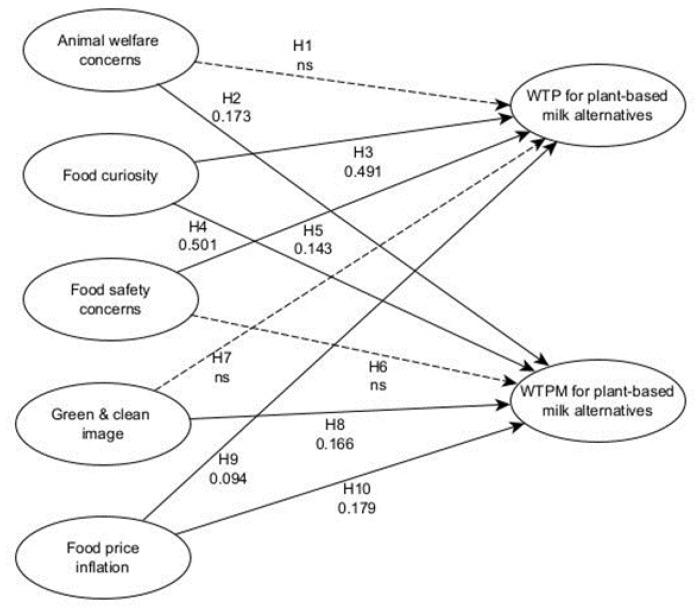
Results. Note: WTP: Willingness to pay; WTPM: Willingness to pay more.

**Table 1 foods-12-01277-t001:** Sample description.

	Freq	%	2019 Census %
Age			
18 to 24	19	3.9	12
25 to 34	192	39.5	18
35 to 44	154	31.7	16
45 to 54	50	10.3	16
55 to 64	51	10.5	17
65 and higher	20	4.1	21
Total	486	100	100
Education			
Failed to finish high school	3	0.6	11
Finished high school	52	10.7	27
Attended university	50	10.3	20
Bachelor’s degree	263	54.1	29
Postgraduate degree	118	24.3	13
Total	486	100	100
Annual Income (Household)			
$0 to under $25 k	74	15.2	18
$25 k to under $50 k	140	28.8	20
$50 k to under $75 k	140	28.8	18
$75 k to under $100 k	92	18.9	13
$100,000 or higher	40	8.2	31
Total	486	100	100
Gender			
Male	258	53.1	49
Female	228	46.9	51
Total	486	100	100
Region			
Northeast	105	21.6	17
South	230	47.3	38
Midwest	86	17.7	21
West	65	13.4	24
Total	486	100	100

Note: 2019 Census %: Percentage of each demographic category in the latest U.S. census.

**Table 2 foods-12-01277-t002:** Item/scale factor loadings, scale reliabilities, and scale convergent validity.

Scales and Items	Factor Loadings	Cronbach’s Alpha	Composite Reliability	Average Variance Extracted
Animal Welfare Concerns		0.89	0.912	0.598
I am highly concerned about animal welfare and factory farming.	0.856			
I do not purchase products where the production process caused animals to suffer.	0.718			
I am concerned about whether the animals were treated humanely and ethically throughout their lives.	0.766			
I am concerned about whether the animals were given adequate food and sanitation.	0.712			
I am concerned about whether the animals were raised as freely and naturally as possible.	0.818			
Plant-based milk alternatives will increase the number of happy animals on earth.	0.757			
The existence of plant-based alternative milk will improve animal welfare conditions.	0.776			
Food Curiosity		0.892	0.917	0.649
Trying new plant-based milk alternatives, such as pea and hemp milk, is very satisfying.	0.885			
Trying new plant-based milk alternatives, such as pea and hemp milk, is an exciting distraction.	0.860			
I am curious to explore new plant-based milk alternatives, such as pea and hemp milk	0.872			
I would describe myself as a foodie.	0.702			
I try and buy new food items when they are available on the market.	0.721			
I like to learn about and try everything related to food.	0.775			
Food Safety		0.815	0.877	0.64
I believe that plant-based milk alternatives are safe.	0.818			
I believe that plant-based milk alternatives are hygienic.	0.806			
I believe that plant-based milk alternatives are clean.	0.791			
I feel that plant-based milk alternatives contain no chemical residues.	0.785			
Green and Clean Image		0.900	0.93	0.77
Plant-based milk alternatives can aid environmental sustainability.	0.895			
Plant-based milk alternatives can help to preserve the environment.	0.884			
Plant-based milk alternatives can help to reduce environmental pollution.	0.893			
Plant-based milk alternatives can help to reduce the use/waste of water.	0.836			
Impact of Food Price Inflation		0.840	0.88	0.596
Due to food price inflation, my food shopping behaviour has changed to include more basic products.	0.677			
Due to food price inflation, my shopping behaviour has changed to include more bulk food.	0.77			
Price increases make me feel threatened.	0.73			
Shortages in food products have led me to competitive and/or panic-buying behaviour.	0.841			
Not having substitute or alternative products makes me anxious.	0.829			
WTP Milk Alternatives (I am willing to buy…)		0.91	0.927	0.615
Almond milk.	0.678			
Rice milk.	0.825			
Soy milk.	0.763			
Coconut milk.	0.738			
Hemp milk.	0.799			
Oat milk.	0.774			
Peanut milk.	0.844			
Pea milk.	0.837			
WTPM Milk Alternatives (I am willing to pay more for…)		0.959	0.965	0.777
Almond milk.	0.838			
Rice milk.	0.911			
Soy milk.	0.866			
Coconut milk.	0.851			
Hemp milk.	0.881			
Oat milk.	0.876			
Peanut milk.	0.907			
Pea milk.	0.917			

Note: WTP: Willingness to pay; WTPM: Willingness to pay more.

**Table 3 foods-12-01277-t003:** Discriminant validity across scales.

Fornell–Larcker Criterion	A	B	C	D	E	F	G
(A) Animal Welfare Concern	0.773						
(B) Food Curiosity	0.573	0.806					
(C) Food Safety	0.551	0.499	0.800				
(D) Green and Clean Image	0.658	0.593	0.586	0.877			
(E) Impact of Food Price Inflation	0.237	0.390	0.145	0.241	0.772		
(F) WTP Milk Alternatives	0.538	0.700	0.500	0.530	0.348	0.784	
(G) WTPM Milk Alternatives	0.581	0.74	0.412	0.587	0.447	0.777	0.881
Heterotrait–Monotrait Ratio	A	B	C	D	E	F	G
(A) Animal Welfare Concern							
(B) Food Curiosity	0.619						
(C) Food Safety	0.622	0.570					
(D) Green and Clean Image	0.689	0.651	0.667				
(E) Impact of Food Price Inflation	0.256	0.383	0.154	0.245			
(F) WTP Milk Alternatives	0.571	0.750	0.585	0.584	0.348		
(G) WTPM Milk Alternatives	0.595	0.783	0.447	0.632	0.440	0.821	

Note: WTP: Willingness to pay; WTPM: Willingness to pay more.

**Table 4 foods-12-01277-t004:** Results of hypothesis testing.

Hypothesized Relationship	Coefficient	T Stat	*p*-Value
H1: Animal Welfare Concerns -> WTP Milk Alternatives	0.119	1.851	0.064
H2: Animal Welfare Concerns -> WTPM Milk Alternatives	0.173	3.071	0.002
H3: Food Curiosity -> WTP Milk Alternatives	0.491	7.935	0.000
H4: Food Curiosity -> WTPM Milk Alternatives	0.501	10.755	0.000
H5: Food Safety -> WTP Milk Alternatives	0.143	2.556	0.011
H6: Food Safety -> WTPM Milk Alternatives	−0.056	1.296	0.195
H7: Green and Clean Image -> WTP Milk Alternatives	0.055	0.903	0.367
H8: Green and Clean Image -> WTPM Milk Alternatives	0.166	3.132	0.002
H9: Impact of Food Price Inflation -> WTP Milk Alternatives	0.094	2.019	0.044
H10: Impact of Food Price Inflation -> WTPM Milk Alternatives	0.179	4.449	0.000
Note: Bold print indicates *p*-value < 0.05			

Note: WTP: Willingness to pay; WTPM: Willingness to pay more.

## Data Availability

The data presented in this study are available upon request from the corresponding author.

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
