# Peer review of "“Got Milk Alternatives?” Understanding Key Factors Determining U.S. Consumers’ Willingness to Pay for Plant-Based Milk Alternatives"

_foods, 2023, doi:10.3390/foods12061277_

Round 1

Reviewer 1 Report

The manuscript by Rombach et al presents some interesting recommendations for marketing and acceptance of plant-based milk alternatives. The language is clear and easy to understand. The hypothesis is sound and clear.

My detail observations are as follows-

       i.          The data collection of food curiosity is the novelty of the present work.

     ii.          Abstract: may be rearranged to: plant-based milk alternatives, animal welfare, food curiosity, PLS-SEM, may also include climate impact/ sustainability

   iii.          Introduction: well written with the sound background information needed to undertake this work. As the WTP and WTPM is very crucial for understanding the market dynamics needed for the further commercialization and popularization of these plant-based milk alternatives.

   iv.          The conceptual model is well explained. Fig 1 excellent depiction of various aspects of the present study.

     v.          The survey sample size (500-14=486) is sufficient and the pre-testing of surveying methods would reduce the possible errors.

   vi.          Table 1: 2019 census refers to the percentage of that group in the overall population? Please add clarity on this.

  vii.          Pea and hemp are the two most commonly used plant-based milk alternatives, and included in your survey under Food curiosity?

Author Response

Reviewer #1:

(x) English language and style are fine/minor spell check required 

A native English speaker has gone through the manuscript and corrected spelling and grammatical errors.

The manuscript by Rombach et al presents some interesting recommendations for marketing and acceptance of plant-based milk alternatives. The language is clear and easy to understand. The hypothesis is sound and clear.

Thank you

My detail observations are as follows-

  1. The data collection of food curiosity is the novelty of the present work.

Thank you.

  1. Abstract: may be rearranged to: plant-based milk alternatives, animal welfare, food curiosity, PLS-SEM, may also include climate impact/ sustainability

Thank you. We have added the key words in the suggested order and added the climate impact and sustainability as suggested.

iii.          Introduction: well written with the sound background information needed to undertake this work. As the WTP and WTPM is very crucial for understanding the market dynamics needed for the further commercialization and popularization of these plant-based milk alternatives.

Thank you.

  1. The conceptual model is well explained. Fig 1 excellent depiction of various aspects of the present study.

Thank you

  1. The survey sample size (500-14=486) is sufficient and the pre-testing of surveying methods would reduce the possible errors.

Thank you for this comment. The questionnaire was pre-tested with 15 workers on Amazon Turk, as discussed in the manuscript. The pre-testing allowed us to make modifications to the survey instrument. These included phrasing of survey questions and scales for improved clarity and allowed us to more accurately estimate the completion time. Pre-testing is crucial at it tends to limit respondent frustration.

  1. Table 1: 2019 census refers to the percentage of that group in the overall population? Please add clarity on this.

We included the following Table 1 note for clarification: Note: US Census %: Percentage of each demographic group in the latest US census.

vii.          Pea and hemp are the two most commonly used plant-based milk alternatives, and included in your survey under Food curiosity?

Yes, we used pea and hemp milk alternatives in some of our food curiosity questions. This was done because pea and hemp are much less popular than the most widely consumed almond, oat, and soy milk alternatives. Following data from Statista and Food Navigator USA, pea milk has a small market share and hemp is too small to be mentioned. In addition, we chose pea as a plant-based milk because consumers may have read about it or had seen on a specialty supermarket shelf, and hemp because consumers were probably not familiar with it and may not have seen it in food retailers. However, we appreciate your comment and have included  a discussion in the limitations and future studies sections. We also included additional market information in the introduction.

Reviewer 2 Report

This manuscript is very interesting, and I believe that this study is valuable for researchers in the field of plant-based milk analogue.

Strength of this manuscript is that authors provided consumers' willingness to pay for the milk alternatives which is a crucial information to develop or to advance the milk alternative products.

Definetely this study is original and novel.

Introduction was informative, and the authors suggested detail hypotheses with logical backgrounds.

Experiment was well designed to suggest the main objective of this study, and results were written by reasonable explanations.

Conclusion highlighted core findings the content briefly and suggested additional research.

The only part that the authors need to reinforce is discussion: The discussion os lacking in explanation of the results, and mainly repeated the background and summary of the results. As suggested in introduction, it would be good approach to compare the results of consumers' willingness to pay for with other plant-based analogue products such as meat analogue. And if possible, it would be necessary to explain in which factors consumers behavior of milk alternatives differ from any other plant-based products.

Author Response

Reviewer 2

This manuscript is very interesting, and I believe that this study is valuable for researchers in the field of plant-based milk analogue. Strength of this manuscript is that authors provided consumers' willingness to pay for the milk alternatives which is a crucial information to develop or to advance the milk alternative products. Definetely this study is original and novel.

Thank you

Introduction was informative, and the authors suggested detail hypotheses with logical backgrounds. Experiment was well designed to suggest the main objective of this study, and results were written by reasonable explanations. Conclusion highlighted core findings the content briefly and suggested additional research.

Thank you!

The only part that the authors need to reinforce is discussion: The discussion os lacking in explanation of the results, and mainly repeated the background and summary of the results. As suggested in introduction, it would be good approach to compare the results of consumers' willingness to pay for with other plant-based analogue products such as meat analogue. And if possible, it would be necessary to explain in which factors consumers behavior of milk alternatives differ from any other plant-based products.

Thank you for these comments. We have expanded the discussion section and have included literature targeting plant food alternatives.